# Enhancing the Photovoltaic Performance of Perovskite Solar Cells Using Plasmonic Au@Pt@Au Core-Shell Nanoparticles

**DOI:** 10.3390/nano9091263

**Published:** 2019-09-05

**Authors:** Bao Wang, Xiangyu Zhu, Shuhan Li, Mengwei Chen, Nan Liu, Hao Yang, Meiqing Ran, Haifei Lu, Yingping Yang

**Affiliations:** 1Department of Physics, School of Science, Wuhan University of Technology, 205 Luoshi Road, Wuhan 430070, China (B.W.) (X.Z.) (M.C.) (N.L.) (H.Y.) (M.R.) (H.L.); 2Wuhan National Laboratory for Optoelectronics, Huazhong University of Science and Technology, 1037 Luoyu Road, Wuhan 430074, China

**Keywords:** Au@Pt@Au core-shell nanoparticles, mesoporous TiO_2_, perovskite solar cells

## Abstract

Au@Pt@Au core-shell nanoparticles, synthesized through chemical reduction, are utilized to improve the photoelectric performance of perovskite solar cells (PSCs) in which carbon films are used as the counter electrode, and the hole-transporting layer is not used. After a series of experiments, these Au@Pt@Au core-shell nanoparticles are optimized and demonstrate outstanding optical and electrical properties due to their local surface plasmon resonance and scattering effects. PSC devices containing 1 wt.% Au@Pt@Au core-shell nanoparticles have the highest efficiency; this is attributable to their significant light trapping and utilization capabilities, which are the result of the distinctive structure of the nanoparticles. The power conversion efficiency of PSCs, with an optimal content of plasmonic nanoparticles (1 wt.%), increased 8.1%, compared to normal PSCs, which was from 12.4% to 13.4%; their short-circuit current density also increased by 5.4%, from 20.5 mA·cm^−2^ to 21.6 mA·cm^−2^. The open-circuit voltages remaining are essentially unchanged. When the number of Au@Pt@Au core-shell nanoparticles in the mesoporous TiO_2_ layer increases, the photovoltaic parameters of the former shows a downward trend due to the recombination of electrons and holes, as well as the decrease in electron transporting pathways.

## 1. Introduction

With the increase of energy demands and the decrease in available fossil fuels worldwide, the development of sustainable energy is one of the most urgent tasks for mankind. Perovskite solar cells (PSCs) are expected to be efficient and low-cost photovoltaic devices for sustainable energy. This is due to their prominent optoelectronic properties, including strong light absorption capacity, long carrier transport length, and low carrier recombination loss [1,2,3]. As a result, since their first use in dye-sensitized solar cells (DSSCs) in 2009, PSCs have attracted a great deal of enthusiasm from researchers worldwide [4,5,6,7,8,9,10,11,12,13]. At present, the maximum photoelectric conversion efficiency of PSC devices has climbed from 3.9% to more than 24%, which is close to the maximum efficiency of single-junction silicon-based photovoltaic devices. Furthermore, PSC devices can be prepared using solution methods with low processing costs and low working temperatures [6,14,15]. The convenience and low cost of fabricating PSC devices makes the commercialization of them possible. 

PSCs are usually expressed using the general-duty expression ABX_3_, in which the A-site is usually an organic cation or metal cation such as CH_3_NH_3_^+^, [(NH_2_)_2_CH]^+^, or Cs^+^; the B-site is generally Pb^2+^; and X represents a halide ion [14,16]. The toxicity of PSCs primarily comes from Pb, a carcinogenic heavy metal and one of the earliest metals used by humans [17,18,19]. The Pb content in PSCs is an issue that is currently restricting their wider commercialization, and decreasing the required amount of Pb to operate PSCs remains a challenge. To this end, some researchers have managed to replace the lead in PSCs with other metallic elements such as Sn, Ge, Bi, or In [9,20,21]. Nevertheless, the photoelectric properties and stability of these low-lead PSC devices are less favorable than conventional devices that use lead. This may be attributed to changes in the lattice arrangement and energy level structure of the perovskite materials after the introduction of other metal cations [11,18,21,22,23].

One effective approach to reduce the lead content of PSC devices, while maintaining their good photoelectric performance, is to introduce metal nanoparticles (NPs) into them. Metal NPs could limit resonant photons and form intense near-field electromagnetic fields, thereby greatly enhancing their light absorption and scattering properties [24,25,26,27]. Light harvesting could be enhanced by a light management foil, TiO_2_ sub-microsphere films, natural random nano-texturing, or bio-inspired nanostructured back electrodes [28,29,30,31]. However, using metal NPs is relatively easier than these light management approaches because of the convenient synthesis process of the former. Moreover, the optical properties of metal NPs could be improved by changing their shape, size, structure, and surrounding environment, suggesting that they have great potential for application in thin film photoelectric devices [32]. Nevertheless, when metal NPs are loaded on the PSCs, the charge recombination and the increased interface roughness are both prominent problems. By utilizing the local surface plasmon resonance (LSPR, Jakarta, Indonesia) and the scattering effect of metal NPs, the ability of PSC absorption layers to absorb light can be greatly enhanced and the thickness of a perovskite thin layer can be indirectly reduced to ensure a high power conversion efficiency (PCE) [33]. Therefore, using metal NPs in PSC devices is a feasible strategy for reducing their lead content while maintaining device performance.

According to our previous studies, because of their LSPR and scattering effects, metal NPs could have a positive impact in hole-conductor-free and carbon counter electrode-based PSCs [34,35,36]. Noble metal core-shell NPs, such as Au@Pt, Pd@Pt, and Au@Pd, show more favorable optical properties, catalytic activity, and greater stability compared to single noble metal NPs [37,38]. The main methods of preparing noble metal core-shell NPs include: Chemical reduction, vapor deposition, and sol-gel processing. TiO_2_ nanoparticles play a significant role in transmitting the electrons that are usually employed as the electron transport layer. These NPs could both scatter and absorb UV light, thereby providing strong UV shielding. There are significant quantum size effects in the TiO_2_ nanoparticles, and as a result, they demonstrate special photophysical and photochemical properties. According to previous literature [39], the electrical conductivity of the TiO_2_ layer improves with the loading of plasmonic NPs, which is helpful for the charge extraction. And Yuan et al. [40] introduced Au NPs into the electron transport layer, which enhanced the conductivity of the electron transport layer through the injection of hot electron; subsequently, they obtained devices of excellent performance. The authors ascribed the enhancement of the lower series resistance, the faster extraction rates of carriers, and the elevation of the Fermi level of the electron transport layer to the incorporation of Au NPs. In this study, we use chemical reduction to synthesize Au@Pt@Au core-shell NPs in order to combine them with the TiO_2_ film of PSC devices. Based on the photoelectric tests conducted during this study, PSCs with Au@Pt@Au core-shell NPs have a higher photoelectric performance than conventional PSCs.

## 2. Materials and Methods

### 2.1. Materials

Chloroauric acid (HAuCl_4_), sodium citrate (Na_3_C_6_H_5_O_7_), silver nitrate (AgNO_3_), and hexachloroplatinic (IV) acid (H_2_PtCl_6_) were purchased from Sinopsin Group Chemical Reagent Co. Ltd. (Shanghai, China). Ultrapure water was used in the experiments. F-doped tin oxide (FTO) conductive substrate was purchased from Advanced Election Technology Co. Ltd. (Yingkou, China). Titanium diisopropoxide bis, dimethyl formamide (DMF), dimethylsulfoxide (DMSO), and methylbenzene were produced by Sigma–Aldrich. The TiO_2_ sizing agent, PbI_2_, and (CH_3_NH_3_)I were provided by Xi’an Polymer Light Technology Corp (Xi’an, China) and ZrO_2_ and the carbon sizing agent were provided by Shanghai MaterWin New Materials Co. Ltd. (Shanghai, China).

### 2.2. Synthesis of Au@Pt@Au Core-Shell NPs

NPs can be prepared via mechanical grinding, ultraviolet irradiation, chemical reduction, and photochemical methods, among others [41,42,43]. Among them, chemical reduction was the simplest to conduct and had minimal equipment requirements. NPs of different sizes and shapes can be prepared by changing the reaction cases; however, impurities were easily introduced during these reactions. It was therefore necessary to clean the nanoparticles obtained as many times as possible to ensure suitable purity for later use.

In this study, we found a chemical reduction method for preparing Au@Pt@Au NPs. First, an aqueous solution of HAuCl_4_ (2.94 × 10^−4^ M, 50 mL) was brought to a boil while being stirred continuously with a magnetic stirrer. An aqueous solution of Na_3_C_6_H_5_O_7_ (3.88 × 10^−2^ M, 1.25 mL) was then added to the HAuCl_4_ solution. After 20 min, the color of the mixture turned bright red, showing that Au NPs were successfully synthesized. Second, an aqueous solution of AgNO_3_ (5.88 × 10^−3^ M, 3 mL) was added dropwise to the mixture and an aqueous solution of Na_3_C_6_H_5_O_7_ (3.88 × 10^−2^ M, 0.75 mL) was mixed into it immediately. After 1 h, an aqueous solution of H_2_PtCl_6_ (1.95 × 10^−1^ M, 0.08 mL) was added and then the solution was stirred quickly for 20 min. After the colloidal solution had cooled, the solution was then washed with moderate ultrapure water for several cycles. The obtained product was diluted to 50 mL with ultrapure water and brought to a boil. An aqueous solution of AgNO_3_ (5.88 × 10^−3^ M, 2.4 mL) and an aqueous solution of Na_3_C_6_H_5_O_7_ (3.88 × 10^−2^ M, 0.6 mL) were then added to the resulting product and stirred with an intense magnetic stirrer for 1 h. After this, an aqueous solution of HAuCl_4_ (2.94 × 10^−4^ M, 0.3 mL) and an aqueous solution of Na_3_C_6_H_5_O_7_ (3.88 × 10^−2^ M, 0.3 mL) were added simultaneously to the colloidal solution. After this, it was stirred quickly for 20 min and Au@Pt@Au NPs were finally formed. After the reaction solution was cooled down, the resulting solution was washed with ultrapure water for several cycles and dried in a drying cabinet for 24 h.

### 2.3. Cell Fabrication

The basic structure of the PSCs prepared in this study was FTO conductive glass/TiO_2_ dense film/TiO_2_ film/ZrO_2_ film/CH_3_NH_3_PbI_3_ film/carbon film. Before preparing the PSCs, the FTO conductive glass was cleaned and TiO_2_ dense film precursor solution, TiO_2_ film colloidal solution, ZrO_2_ film colloidal solution, and CH_3_NH_3_PbI_3_ precursor solutions were prepared. The FTO conductive glass was sequentially cleaned with ultrapure water, dimethyl ketone, isopropanol alcohol, and an ultrasonic ethanol treatment. The TiO_2_ dense layer prefabricated liquid was obtained by mixing 1 mL titanium diisopropoxide bis with 19 mL ethanol, while the TiO_2_ mesoporous layer colloidal solution and ZrO_2_ mesoporous layer colloidal solution were prepared by adding 2 g ethyl alcohol to 0.5 g TiO_2_ or a ZrO_2_ sizing agent. Then, 1–3 mg Au@Pt@Au NPs were added to the TiO_2_ mesoporous layer colloidal solution, which was ultrasonicated for 30 min and stirred for 48 h. The perovskite precursor solution consisted of 231 mg PbI_2_, 89 mg (CH_3_NH_3_)I, 300 mg DMF, and 78 mg DMSO.

The methods available for the fabrication of PSC devices include: Spin-coating, vapor deposition, and ultrasonic spraying, among others [44,45,46,47,48]. In this study, we utilized a spin-coating method. First, 35 μL of TiO_2_ compact layer prefabricated liquid was deposited on glass using the spin-coating method at a speed of 4000 r/min for 20 s, followed by 30 min of annealing at 500 °C. Then, the first step was repeated for the TiO_2_ and ZrO_2_ mesoporous layer colloidal solutions with Au@Pt@Au or Au NPs. The CH_3_NH_3_PbI_3_ film was prepared in an airtight box filled with nitrogen. First, 35 μL of CH_3_NH_3_PbI_3_ solution was spin-coated on the substrate at 1000 r/min for 10 s and 4000 r/min for 15 s, followed by 10 min of annealing at 100 °C. During this spin-coating, 300 μL methylbenzene was added quickly to the solution in order to improve the quality of the film being formed [45]. Finally, a carbon film was obtained by using a screen-printing board and being heated at 100 °C for 30 min.

### 2.4. Characterization

A transmission electron microscope (TEM; JEOL, Tokyo, Japan) was used to observe the ultrastructure of the NPs. X-ray diffraction (XRD; AXS, Los Angeles, CA, USA) was utilized to investigate the phases of the as-prepared samples, while X-ray photoelectron spectroscopy (XPS, Thermo Fisher Scientific, Waltham, MA, USA) was used to assess the binding energies of the elements in the samples. The absorption curves were collected using a UV-vis spectrophotometer (UV3600, Shimadzu, Kyoto, Japan). Cross-sections of the PSC samples were imaged by using a scanning electron microscope (SEM; JEOL, Tokyo, Japan). The photoelectric properties of the PSCs were evaluated, based on the photocurrent-voltage (J-V) recorded on an electrochemical workstation (ZAHNER-elektrik GmbH and Co. KG, Kronach, Germany) under simulated solar light (Oriel Sol3A, Newport Corporation, Irvine, CA, USA). The measurements were carried out from –1.1 V to short circuit voltage at a scan rate of 150 mV/s under air mass (AM) 1.5 G irradiation (100 mW/cm^2^) in ambient air. Incident photon-to-electron conversion efficiency (IPCE) curves were acquired with a device produced by the Newport Corporation, USA, in order to analyze the photoelectric current of the sample cells under dark conditions in ambient air.

## 3. Results and Discussion

The process of preparing Au@Pt@Au NPs was shown in Figure 1. First, Na_3_C_6_H_5_O_7_ was used to transform Au ions in HAuCl_4_ to Au NPs. Then, an Ag shell was prepared on the surfaces of the Au NPs; this was subsequently replaced by a Pt shell. Finally, another Ag shell was synthesized on the surfaces of the Au@Pt core-shell NPs and was then replaced with several small Au spheres around the Au@Pt core-shell NPs. This yielded Au@Pt@Au core-shell NPs.

To investigate the morphology of these NPs, TEM and high-resolution TEM (HRTEM) were used to determine their size, shape, and structure. As depicted in Figure 2a,c,d, Au NPs, Au@Pt core-shell NPs, and Au@Pt@Au core-shell NPs were scattered uniformly in deionized water according to the bar graphs given in these images. The radii of Au NPs were approximately 15 nm, while those of the Au@Pt core-shell NPs were nearly 18 nm, which indicated that the Pt shells were approximately 3 nm thick. The nucleation sizes of the Au@Pt@Au core-shell NPs were almost equal to those of the Au@Pt core-shell NPs; the difference was that the Au@Pt@Au core-shell NPs were surrounded by several small Au spheres (with radii of approximately 5 nm). Figure 2b shows an enlarged image of an Au NP, in which the 2.36 Å long lattice fringes can be observed clearly and correspond to the Au (111) crystal plane [49]. Several small Au spheres delineated by the red trajectories in Figure 2e were distributed on the surface of an Au@Pt core-shell NP. The three-dimensional model of an Au@Pt@Au core-shell NP in Figure 2f corresponds to the NPs in Figure 2d,e.

Figure 3a shows the UV-visible absorption curves of Au NPs, Au@Pt core-shell NPs, and Au@Pt@Au core-shell NPs dispersed in deionized water. The LSPR resonance absorption peak of Au NPs (r = 15 nm) appeared at approximately 520 nm. The absorption peak of Au@Pt core-shell NPs had a blue shift of approximately 20 nm, compared to the Au NPs, which could be ascribed to the shielding effect of the Pt shell. The blue curve, representing the optical absorption spectrum of the Au@Pt@Au core-shell NPs, had two peaks at 380 nm and 600 nm, respectively. The absorption peak at 380 nm was generated by the inner Au@Pt core-shell NPs, while the peak at 600 nm corresponded to the small, outermost Au spheres. The black line in Figure 3b represents the XRD pattern of the powder obtained by calcining the TiO_2_ mesoporous layer colloidal solution at 500 °C. The diffraction angles observed at 25°, 38°, and 48°, respectively, corresponded to the (101), (004), and (200) crystal planes of anatase phase TiO_2_. However, the red line, depicting the sample mixed with Au@Pt@Au core-shell NPs, was almost identical to the black one. There were no distinct characteristic peaks of Au and Pt in the XRD pattern; this may be because of the low-content Au@Pt@Au core-shell NPs in the TiO_2_ mesoporous layer colloidal solution.

XPS was used to investigate the elemental compositions and chemical states of the mesoporous TiO_2_ samples with or without Au@Pt@Au core-shell NPs. Figure 4a,b revealed the photoelectron energies of the Ti 2p and O 1s of the mesoporous TiO_2_ samples and mesoporous TiO_2_ samples incorporating Au@Pt@Au core-shell NPs. The peaks of the red lines were almost consistent with the black ones, which implied that the chemical states of the Ti and O atoms did not change after the Au@Pt@Au core-shell NPs were incorporated. As depicted in Figure 4c,d, four characteristic peaks were found at 83.5 eV, 87.5 eV, 71.5 eV, and 74.5 eV, which could be indexed to Au0 and Pt0, respectively, which suggested that both Au0 and Pt0 were present in the samples [50,51,52].

Figure 5a,b show the structure and SEM cross-section images of an entire mesoporous PSC. From the bottom of the cell up, the layers are the substrate, TiO_2_ dense film, TiO_2_ film mixed with Au@Pt@Au core-shell NPs, ZrO_2_ film, CH_3_NH_3_PbI_3_ film, and carbon film. The carbon film assembled via silk-screen printing was 30 μm thick, which was far thicker than the other layers. Therefore, the device characterized by SEM did not include the electrode film. The TiO_2_ compact layer was usually 20 nm, but this was not clear enough to be visible in Figure 5b.

To further explore the optical properties of the PSC devices, UV-vis curves were recorded in order to investigate the light absorption capacity of the devices that incorporated different quantities of Au@Pt@Au core-shell NPs. As described in Figure 6a, the intensity of the UV-vis absorption spectra increased gradually, particularly at 400–600 nm, as the Au@Pt@Au NPs load increased. This was consistent with the absorption curve of pure Au@Pt@Au core-shell NPs. Figure 6b shows a schematic of the LSPR and the scattering effects in the PSCs. When the vibrational frequency of the photons matched well with the frequency of the Au@Pt@Au core-shell NPs, an intense local electromagnetic field was generated around the NPs, which could cause bandgap excitation in the nearby TiO_2_ and generate more electron-hole pairs in the cell [25,53,54]. Furthermore, Au@Pt@Au core-shell NPs display excellent scattering because of the outermost, small Au spheres, which could ensure that the light transmitted back and forth is more efficiently utilized by the device. A high temperature would affect the LSPR effect of plasmonic NPs; however, an enhancement seen in the absorbance spectra was related with the LSPR effect and the scattering effect of NPs. Figure 6a shows that the LSPR effect and the scattering effect of NPs still played a role in the TiO_2_ layers after annealing.

The J-V curves and parameters of cells incorporated with different amounts of Au@Pt@Au core-shell NPs are presented in Figure 7 and Figure 8, and Table 1, respectively. These parameters were tested with an electrochemical workstation at 25 °C in air. As the concentration of Au@Pt@Au core-shell NPs varied from 0 wt.% to 1.5 wt.%, both the short-circuit current densities (*J_SC_*) and the PCEs first increased and then decreased. The enhancements could be ascribed to the incorporation of Au@Pt@Au core-shell NPs, which were the centers of strong electromagnetic fields produced by the LSPR effect and scattering centers in the cells, thereby improving the light utilization rate. However, this diminution might be ascribed to the reorganization of electrons and holes on the surface of Au@Pt@Au core-shell NPs, and could also be attributed to the decrease in electron travelling pathways with increased Au@Pt@Au core-shell NP loading [55,56]. Compared to the reference cells, the performance parameters of the devices containing 1 wt.% Au@Pt@Au core-shell NPs improved by 8.1% in terms of PCEs, from 12.4% to 13.4%, and by 5.4% at *J_SC_*, from 20.5 mA·cm^−2^ to 21.6 mA·cm^−2^, respectively, while the open-circuit voltages (*V_OC_*) were essentially unchanged. In addition, there was a slight decline in the fill factors (FFs), which could be attributed to the increase in electronic traps after the incorporation of excess Au@Pt@Au core-shell NPs.

The hysteresis index could be calculated by the following Formula [57]:(1)$$\mathrm{Hysteresis}=\frac{\mathrm{PCE}\left(\mathrm{reverse}\right)-\mathrm{PCE}\left(\mathrm{forward}\right)}{\mathrm{PCE}\left(\mathrm{reverse}\right)},$$

Therefore, according to Formula (1), the hysteresis index, with and without plasmonic NPs, was 0.089 and 0.084, respectively. The result indicated that the hysteresis effect had only changed a little with the loading of plasmonic NPs in these architecture-based PCSs.

Figure 8 describes the box charts for the photovoltaic parameters of devices based on mesoporous TiO_2_ films mixed with 0–2 wt.% Au@Pt@Au core-shell NPs, which indicated that the photoelectric properties of the cells remained steady.

External quantum efficiency spectra were measured to further study the photoelectric conversion capacity of the cells. Figure 9 shows the IPCE spectra and integrated current density curves of the PSC devices mixed with Au@Pt@Au core-shell NPs. The trend in these IPCE spectra were consistent with the J-V characteristics described earlier, which indicated that the PSC devices containing 1 wt.% Au@Pt@Au core-shell NPs had the best light trapping and utilizing capability of all the devices described in this study. When the concentration of Au@Pt@Au core-shell NPs increased past this optimal value, the enhancements from the effects of LSPR and scattering were not enough to resist the decrease caused by the recombination of electron hole pairs near the electron traps. Hence, it was of great importance to control the amount of NPs in order to obtain high-efficiency light harvesters. The integrated current density curves were acquired by integrating the IPCE spectra in the left-hand part of Figure 9. The integral results were slightly lower than the *J_SC_* numbers in Table 1, which may have been influenced by the equipment and the test environment.

## 4. Conclusions

We have adopted a chemical reduction method to prepare Au@Pt@Au core-shell NPs to be mixed with the TiO_2_ mesoporous layer of PSCs. The TEM images, optical absorption spectra, XRD patterns, and XPS spectra resulting from the tests on these modified PSCs were used to characterize the physicochemical properties of the Au@Pt@Au core-shell NPs. Furthermore, SEM cross-section images, UV-vis absorption spectra, J-V characteristics, histograms, and IPCE curves were used to investigate the photoelectric performance of the PSC cells with different concentrations of Au@Pt@Au core-shell NPs. PSC devices containing 1 wt.% Au@Pt@Au core-shell NPs had the best photovoltaic performances, and these were ascribed to the LSPR and scattering effects of the NPs. Nevertheless, when excess Au@Pt@Au core-shell NPs were mixed into the devices—i.e., when Au@Pt@Au core-shell NP loading increased—the efficacy of the photovoltaic parameters decreased due to the reorganization of electrons and holes on the surface of Au@Pt@Au core-shell NPs and the decrease in electron travelling pathways.

## Figures and Tables

**Figure 1 nanomaterials-09-01263-f001:**
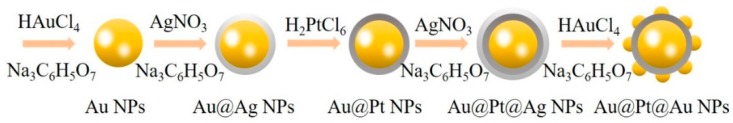
The preparation process picture of Au@Pt@Au core-shell nanoparticles (NPs).

**Figure 2 nanomaterials-09-01263-f002:**
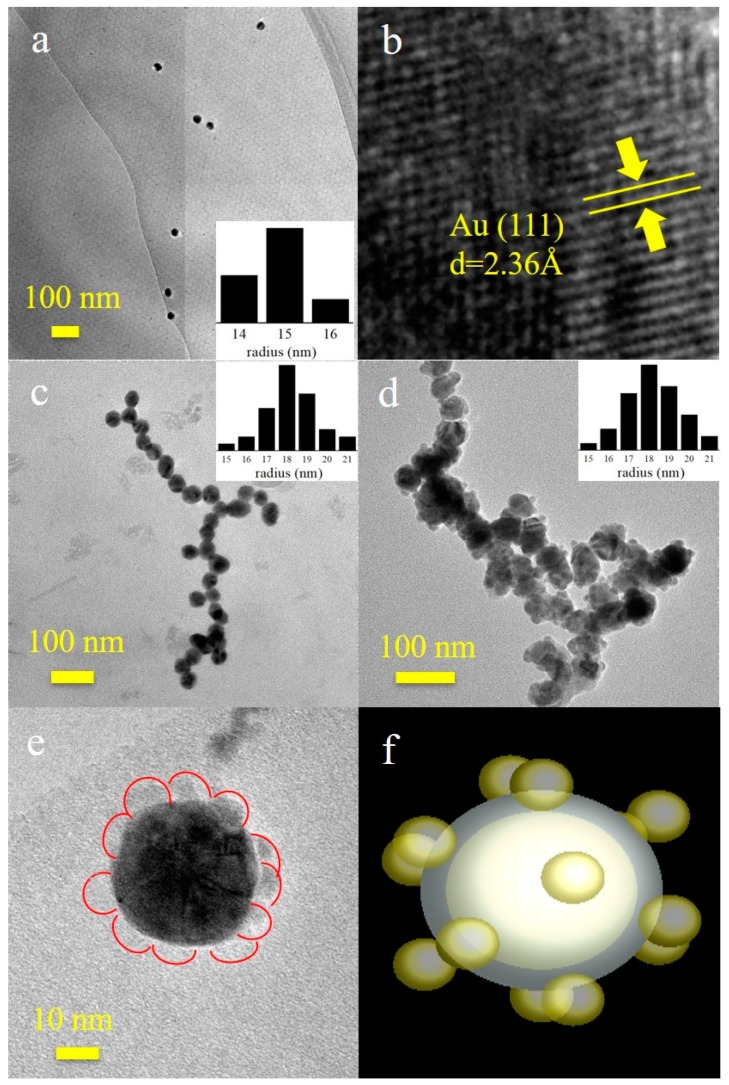
Transmission electron microscope (TEM) images and size distribution results of (**a**) Au NPs, (**c**) Au@Pt core-shell NPs, and (**d**) Au@Pt@Au core-shell NPs; (**b**) enlarged image of an Au NP; (**e**) high-resolution TEM (HRTEM) image; and (**f**) the three-dimensional model of an Au@Pt@Au core-shell NP.

**Figure 3 nanomaterials-09-01263-f003:**
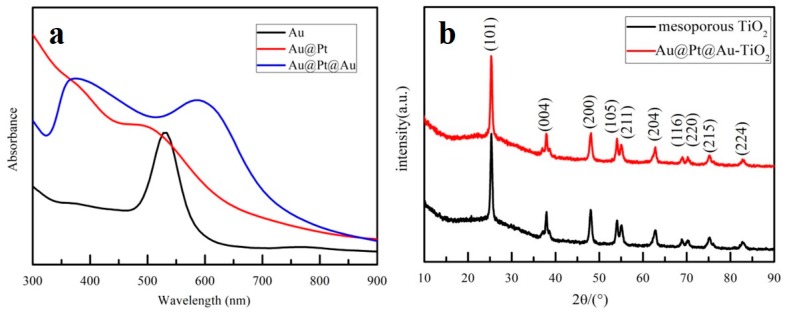
(**a**) Optical absorption curves of Au NPs, Au@Pt core-shell NPs, and Au@Pt@Au core-shell NPs dispersed in deionized water; (**b**) X-ray diffraction (XRD) atlas of TiO_2_ and TiO_2_ incorporated with Au@Pt@Au core-shell NPs.

**Figure 4 nanomaterials-09-01263-f004:**
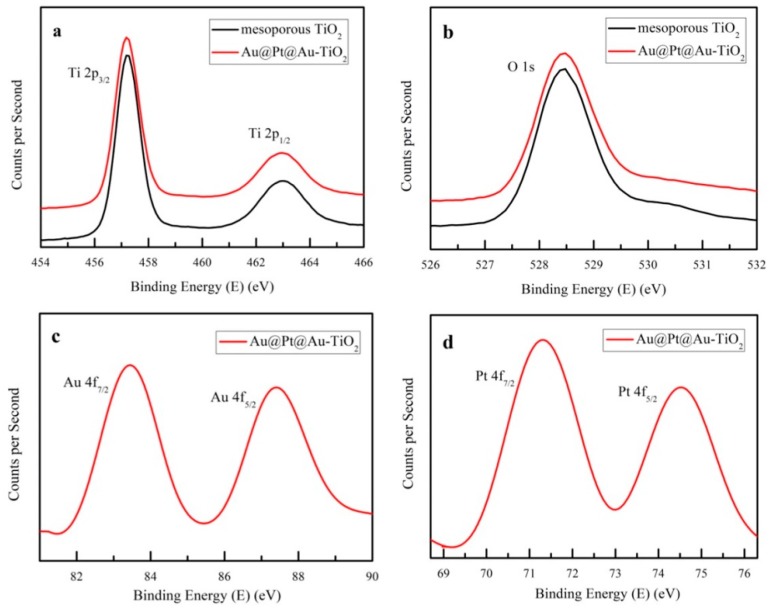
(**a**) Ti 2p, (**b**) O 1s, (**c**) Au 4f, and (**d**) Pt 4f XPS spectra of mesoporous TiO_2_ samples and mesoporous TiO_2_ samples incorporated with Au@Pt@Au core-shell NPs.

**Figure 5 nanomaterials-09-01263-f005:**
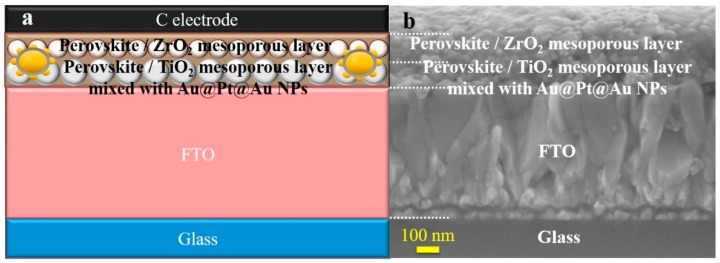
(**a**) The structure drawing of a whole mesoporous perovskite solar cells (PSC); (**b**) a cross image of a sample device.

**Figure 6 nanomaterials-09-01263-f006:**
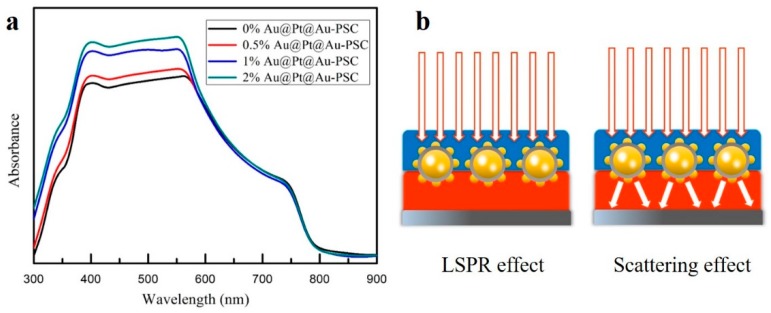
(**a**) UV-vis curves of cells with mesoporous TiO_2_ and mesoporous TiO_2_ incorporated with different amounts of Au@Pt@Au core-shell NPs; (**b**) the schematic diagram of the local surface plasmon resonance (LSPR) effect and scattering effect in PSCs.

**Figure 7 nanomaterials-09-01263-f007:**
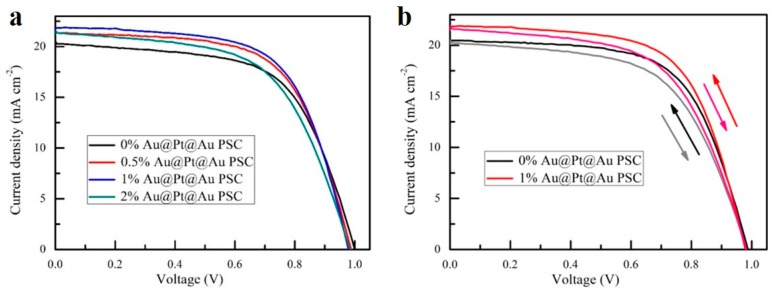
(**a**) J-V characteristics of cells incorporated with different amounts of Au@Pt@Au core-shell NPs under AM 1.5G irradiation (100 mW cm^−2^); (**b**) J-V curves scanned from forward voltage to reverse voltage and scanned in the opposite direction.

**Figure 8 nanomaterials-09-01263-f008:**
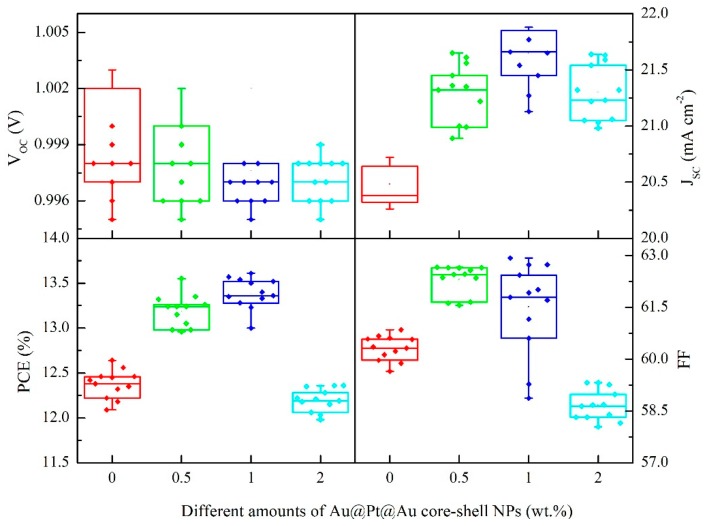
Box charts of photovoltaic arguments for devices incorporated with 0–2 wt.% Au@Pt@Au core-shell NPs under simulated AM 1.5G irradiation (100 mW cm^−2^).

**Figure 9 nanomaterials-09-01263-f009:**
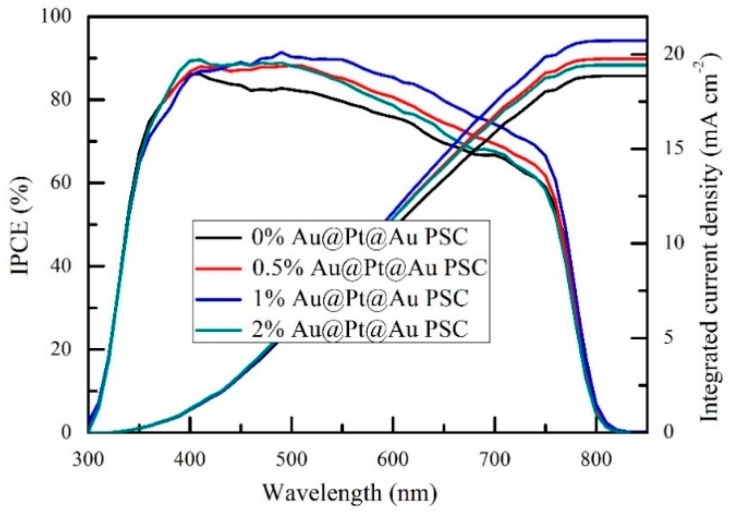
External quantum efficiency spectra and the integrated current density curves of the cells incorporated with different amounts of Au@Pt@Au core-shell NPs.

**Table 1 nanomaterials-09-01263-t001:** Summarized photovoltaic parameters of different cells.

Samples	*V_OC_* (V)	*J_SC_* (mA∙cm^−2^)	FF (%)	PCE (%)
0 wt.% Au@Pt@Au-TiO_2_	1.00 ± 0.01	20.5 ± 0.2	60.3 ± 0.4	12.4 ± 0.2
0.5 wt.% Au@Pt@Au-TiO_2_	1.00 ± 0.01	21.3 ± 0.2	62.3 ± 0.4	13.2 ± 0.2
1 wt.% Au@Pt@Au-TiO_2_	1.00 ± 0.01	21.6 ± 0.3	61.5 ± 1.3	13.4 ± 0.2
2 wt.% Au@Pt@Au-TiO_2_	1.00 ± 0.01	21.3 ± 0.2	58.7 ± 0.5	12.2 ± 0.1

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
