# Peer review of "Enhancing the Photovoltaic Performance of Perovskite Solar Cells Using Plasmonic Au@Pt@Au Core-Shell Nanoparticles"

_nanomaterials, 2019, doi:10.3390/nano9091263_

Round 1
Reviewer 1 Report
The authors demonstrates that Au@Pt@Au core-shell nanoparticles can improve the light scattering and absorption within perovskite layer, increase solar cell performance with proper amount of Au@Pt@Au nanoparticels and reduce the inserted amount of lead contents by reducing the perovskite layer thickness. The author well-wrote and well-explained the novelty and experimental resutls in this manuscript. I recommend this manuscript is suitable for publication in Nanomaterials. I have minor question as below.
1. Author incorprated Au@Pt@Au nanoparticels into mesoporous TiO2. I am now wondering that metal nanoparticles could be sintered by thermal annealing process of TiO2. Generally, metal nanoparticels can be easilty sintered at lower temperature than their melting point because of high surface area. That means LSPR effect of metal nanoparticles could be disappeared after annealing. Please clarify this issue in your manuscript.
Author Response
1. Author incorprated Au@Pt@Au nanoparticels into mesoporous TiO2. I am now wondering that metal nanoparticles could be sintered by thermal annealing process of TiO2. Generally, metal nanoparticels can be easilty sintered at lower temperature than their melting point because of high surface area. That means LSPR effect of metal nanoparticles could be disappeared after annealing. Please clarify this issue in your manuscript.
It’s a good suggestion. Indeed, high temperature would affect the LSPR effect of plasmonic NPs. And we haven’t considered about this problem. However, we could view the enhancement in the absorbance spectra, which was related with the LSPE effect and scattering effect of NPs. Also, we have measured the absorbance spectra of TiO2 layers with different amounts of Au@Pt@Au core-shell nanoparticles. The figure was displayed as below. From this figure, we could get this point that the LSPE effect and scattering effect of NPs still played a role in the TiO2 layers after annealing.
And we have revised our manuscript in relevant place.

Reviewer 2 Report
The authors demonstrate the use of metal nanoparticles in the meso-porous TiO2 film, to enhance the PCE of perovskite solar cells. Though the topic in general is relevant, but the quality of this work is not up to the standard of Nanomaterials, in its current form.
Following are some major concerns:
Title too long. In general sentences are too long and English needs to be improved throught out the text.
Abstract is too general and provides no specific information
Given that the PSC now have reported efficiencies of over 24%, author do not discuss/demonstrate the use of such nanoparticles with other high performance perovskite materials. Also the reference PCE of devices with no nanoparticles in mesoporous TiO2 are far lower than literature values.
Authors did not discussed how the interfacial energetics in devices change with the introduction of nanoparticles. For example, the WF of TiO2 with nanoparticles or the lateral distribution in surface potential. UPS, KPFM etc?
Table 1: "arguments" is the wrong word used
Electrical properties of TiO2 layer is not included/discussed to understand the devices trends.
In conclusions, Authors are not specific about the mechanism that contributes to enhancement of device PCE
Given the reproducibility issues in PSCs, authors must be scientifically more explicit if the differences in the devices parameters in table 1 are significant.
How does the Hysterisis change with introduction of nanoparticles in TiO2?
Author Response
1. Title too long. In general sentences are too long and English needs to be improved throught out the text.
It’s a good suggestion. We have revised the title from ‘Enhanced photovoltaic performance of perovskite solar cells using plasmonic Au@Pt@Au core-shell nanoparticles deposited in mesoporous TiO2 layer’ to ‘Enhanced photovoltaic performance of perovskite solar cells using plasmonic Au@Pt@Au core-shell nanoparticles’.
Thanks for your advice and the English writing of the manuscript has been carefully edited by a native English speaker. CERTIFICATE OF ENGLISH EDITING is appended below:
2. Abstract is too general and provides no specific information
It’s a good suggestion. We have revised our abstract carefully. The detailed description are as follows:
Perovskite solar cell devices containing 1 wt. % Au@Pt@Au core-shell nanoparticles performed the highest efficiency; this was ascribed to the prominent light trapping and utilization capability because of the distinctive structure of the nanoparticles. Compared with normal PSCs, PSCs with an optimal content of plasmonic nanoparticles (1 wt%) enhanced 9.00% in terms of power conversion efficiency, from 12.45% to 13.57% and 6.38% at short-circuit current density, from 20.54 mA·cm-2 to 21.85 mA·cm-2 respectively, while the open-circuit voltages were essentially unchanged.
3. Given that the PSC now have reported efficiencies of over 24%, author do not discuss/demonstrate the use of such nanoparticles with other high-performance perovskite materials. Also, the reference PCE of devices with no nanoparticles in mesoporous TiO2 are far lower than literature values.
It’s a good suggestion. In the FTO/c-TiO2/m-TiO2/ZrO2/perovskite/carbon architecture-based PCSs, we didn’t adopt expensive noble metal as electrode and sprio-OMeTAD as HTL. Therefore, the cost of fabricating the devices is relatively low, which is more commercialization friendly. Besides, the use of carbon back electrodes makes it possible to product large-area PSCs [1]. However, without the metal electrode and HTL, the PCEs of the devices were relatively low.
According to Han’s reports [1-2], the PCE of the devices with the same cell configuration was about 13%, which was very consistent with our report.
[1] Mei, A.; Li, X.; Liu, L.; et al. A hole-conductor-free, fully printable mesoscopic perovskite solar cell with high stability. Science 2014, 345, 295-298. [https://doi.org/10.1126/science.1254763]
[2] Pei, J.; Timothy, W.; Noel, W.; et al. Fully printable perovskite solar cells with highly-conductive, low-temperature, perovskite-compatible carbon electrode. Carbon 2018, 129, 830-836. [https://doi.org/10.1016/j.carbon.2017.09.008]
4. Authors did not discuss how the interfacial energetics in devices change with the introduction of nanoparticles. For example, the WF of TiO2 with nanoparticles or the lateral distribution in surface potential. UPS, KPFM etc.?
It’s a good suggestion. We are sorry for that we don’t have sufficient condition to do relevant measurements, like UPS, KPFM.
5. Table 1: "arguments" is the wrong word used
Thank you for pointing out our error. We have revised the word from ‘arguments’ to ‘parameters’.
6. Electrical properties of TiO2 layer is not included/discussed to understand the devices trends.
It’s a good suggestion. We have added some information about the electrical properties of TiO2 layer in the manuscript. The detailed description are as follows:
TiO2 nanoparticles played a significant role in the transmission of electrons, which were usually employed as the electron transport layer. TiO2 nanoparticles could not only scatter uv light, but also absorbs uv light, which provided strong uv shielding. There are significant quantum size effects in the TiO2 nanoparticles. Therefore, TiO2 nanoparticles showed up special photophysical and photochemical properties.
7. In conclusions, Authors are not specific about the mechanism that contributes to enhancement of device PCE
It’s a good suggestion. Indeed, we didn’t give out the specific mechanism that contributes to enhancement of device PCE. We ascribed the enhancement of PCE to the absorbance enhancement, which was related with the absorbance and scattering of plasmonic NPs.
8. Given the reproducibility issues in PSCs, authors must be scientifically more explicit if the differences in the devices parameters in table 1 are significant.
It’s a good suggestion. We have revised Figure 8, in which we could get a clear view about the photovoltaic parameters of PSC devices. The detailed description are as follows:
Figure 8. Box charts of photovoltaic arguments for devices incorporated with 0-2 wt.% Au@Pt@Au core-shell NPs under simulated AM 1.5G irradiation (100 mW cm-2).
9. How does the Hysteresis change with introduction of nanoparticles in TiO2?
It’s a good suggestion. Certainly, PSCs usually suffer from a hysteresis effect in I-V measurements, which was related with measurement settings [1-3]. In the FTO/c-TiO2/m-TiO2/ZrO2/perovskite/carbon architecture-based PCSs, the properties of the c-TiO2 layer and corresponding interfaces significantly affect the I-V hysteresis [4]. In this work, we had observed slight hysteresis change with introduction of nanoparticles in TiO2. The detailed description are as follows:
[1] Nemnes, G.A.; Besleaga, C.; Tomulescu, A.G.; Pintilie, I.; Pintilie, L.; Torfason, K.; Manolescu, A. Dynamic electrical behavior of halide perovskite based solar cells. Sol. Energy Mater. Sol. Cells 2017, 159, 197-203.
[2] Nemnes, G.A.; Besleaga, C.; Stancu, V.; Dogaru, D.E.; Leonat, L.N.; Pintilie, L.; Torfason, K.; Ilkov, M.; Manolescu, A. Normal and Inverted Hysteresis in Perovskite Solar Cells. J. Phys. Chem. C 2017, 121, 11207−11214.
[3] Snaith, H.J.; Abate, A.; Ball, J.M.; Eperon, G.E.; Leijtens, T.; Noel, N.K.; Stranks, S.D.; Wang, J.T.W.; Wojciechowski, K.; Zhang W. Anomalous Hysteresis in Perovskite Solar Cells. J. Phys. Chem. Lett. 2014, 5, 1511−1515.
[4] Rong, Y.; Hu, Y.; Ravishankar, S.; Liu, H.; Hou, X.; Sheng, Y.; Mei, A.; Wang, Q.; Li, D.; Xu, M.; Bisquert, J.; Han H. Tunable hysteresis effect for perovskite solar cells. Energy Environ. Sci. 2017, 10.

Reviewer 3 Report
This paper reports the incorporation of Au@Pt@Au core-shell nanoparticles (NPs) in the mesoporous electron transport layer (ETL) of perovskite solar cells (PSCs) using a carbon coating as back electrode instead of a hole transport layer (HTL)/metal mirror structure. It shows an improvement in the power conversion efficiency (PCE) with respect to a reference cell without NPs for an optimal concentration of NPs (1 wt%). This improvement is mostly due to a short circuit current (Jsc) increase, which the authors attribute to near-field enhancement and scattering by the NPs.
This paper might be publishable; however, in my opinion it must be improved on several points detailed hereafter and reviewed again before a decision can be taken.
Lines 54-62: Please comment on the advantages/drawbacks of introducing NPs in the ETL of PSCs with respect to other approaches involving scattering structures. Light scattering can be achieved by structuring the front glass surface, introducing high index dielectric nanostructures within the PSC, or structuring the back electrode. For instance, consider the following references:
Jost, ACS Photon. 4, 1232-1239 (2017); https://doi.org/10.1021/acsphotonics.7b00138
Huang, ACS Appl. Mater. & Interfaces 8, 8162-8167 (2016); https://doi.org/10.1021/acsami.5b08421
Zhang, ACS Photon. 5, 2243-2250 (2018); https://doi.org/10.1021/acsphotonics.8b00099
Wei, Adv. Energy Mater. 1700492 (2017); https://doi.org/10.1002/aenm.201700492
Lines 64-66: Please explain the motivation to use a carbon back electrode without HTL. This is not the best configuration to achieve a high PCE.
How does the best device of this work compare in terms of PCE with other devices with the same cell configuration (FTO/meso-ETL/MAPI/C) reported in the literature?
Line 108: Please explain why the ZrO2 layer is deposited on top of the TiO2 one.
Line 137: Please explain how the absorbance was measured. With or without integrating sphere? Using the relation Absorbance = 1 – Reflectance – Transmittance?
Line 142: Please provide more details about the PV & EQE measurements (forward/reverse scan, scan rate, light/voltage bias…).
Figure 2: Do the authors have at hand a better TEM image showing more clearly than in (e) the Pt shell and the outer Au NPs? Please change “size” to “radius”
Line 166: Do the authors mean “radius” or “diameter”?
Figure 3: How was absorption measured? With or without integrating sphere? Using the relation Absorbance = 1 – Reflectance – Transmittance?
Line 183: Please provide a more solid evidence to the fact that the 380 nm and 800 nm resonances correspond to the inner and outer Au NPs: simulations or reference.
Figure 5: This figure is misleading, because the drawing on the left has a different vertical scale as the SEM image on the right. Please provide a better representation.
Figure 6: Please explain how the absorbance spectra were measured. On devices without carbon coating? With or without integrating sphere? Using the relation Absorbance = 1 – Reflectance – Transmittance? If only the transmittance was measured please show the transmittance spectrum instead of absorbance.
Also please provide the spectra in %, not arbitrary units.
Lines 222-226: Adding the NPs increases the device absorbance in the 300 – 550 nm range and not at all at longer wavelengths (figure 6a). In contrast, the plasmon resonances of the NPs extend from 300 to more than 700 nm (figure 3a). If the plasmon resonances extend up to more than 700 nm and the NPs have a significant contribution to the device absorbance, then one might expect an improvement of the device absorbance at longer wavelengths than 550 nm upon adding the NPs. Why is it not observed? Please explain.
Line 231: “The NPs can display excellent scattering because of the outermost small Au spheres”. This is not trivial. Please provide a reference or simulations to support this fact.
Table 1: Is there a way to estimate the volume fraction of NPs for a given wt%? What matters in optics is the volume fraction of absorbers/scatterers. This is needed to assess how much the NPs can affect the device absorbance.
Figure 8: This figure is unclear. Please modify it. For example, use the same representation as in the front image of this work: https://doi.org/10.1039/C9TA01744E.
If possible include also the data for the reference cell (no NPs) and the cells with other NP wt% (not only 1%).
Figure 9: Why does EQE increase over the whole 300-800 nm spectral range, while the absorbance in figure 6 only increases in the 300 – 550 nm range? This does not seem very consistent with the fact that only optical effects affect EQE & Jsc. Please explain.
I also wonder if the NPs might have a positive effect on charge extraction in short-circuit conditions.
Author Response
1. Lines 54-62: Please comment on the advantages/drawbacks of introducing NPs in the ETL of PSCs with respect to other approaches involving scattering structures. Light scattering can be achieved by structuring the front glass surface, introducing high index dielectric nanostructures within the PSC, or structuring the back electrode. For instance, consider the following references:
Jost, ACS Photon. 4, 1232-1239 (2017); https://doi.org/10.1021/acsphotonics.7b00138
Huang, ACS Appl. Mater. & Interfaces 8, 8162-8167 (2016);
https://doi.org/10.1021/acsami.5b08421
Zhang, ACS Photon. 5, 2243-2250 (2018);
https://doi.org/10.1021/acsphotonics.8b00099
Wei, Adv. Energy Mater. 1700492 (2017); https://doi.org/10.1002/aenm.201700492
We have carefully investigated the similar research and corrected our manuscript. The detailed corrections are as followed:
Light harvesting could be enhanced by a light management foil, TiO2 sub-microsphere films, natural random nano-texturing, or a bioinspired nanostructured back electrode [28-31]. However, compared with these light management approaches, the method of using metal NPs was relatively easy because of the convenient synthesis process. Moreover, the optical properties of metal NPs could be improved by changing their shape, size, structure, and surrounding environment, suggesting that metal NPs have great application potential in thin film photoelectric devices [32]. Nevertheless, with the loading of metal NPs, the charge recombination and the increased interface roughness were both the prominent problems.
[28] Jost, M.; Albrecht, S.; Kegelmann, L.; et al. Efficient light management by textured nanoimprinted layers for perovskite solar cells. ACS Photonics 2017, 4, 1232-1239. [https://doi.org/10.1021/acsphotonics.7b00138]
[29] Yang, H.; Jun, Z.; Yong, D.; et al. TiO2 sub-microsphere film as scaffold layer for efficient perovskite solar cells. ACS Appl. Mater. 2016, 1-13. [https://doi.org/10.1021/acsami.5b08421]
[30] Hui, Z.; Mariia, K.; Johann, O.; et al. Natural random nano-texturing of the Au interface for light backscattering enhanced performance in perovskite solar cells. ACS Photonics 2018, 1-28. [https://doi.org/10.1021/acsphotonics.8b00099]
[31] Jian, W.; Rui-Peng, X.; Yan-Qing, L.; et al. Enhanced light harvesting in perovskite solar cells by a bioinspired nanostructured back electrode. Communication 2017, 1700492. [https://doi.org/10.1002/aenm.201700492]
2. Lines 64-66: Please explain the motivation to use a carbon back electrode without HTL. This is not the best configuration to achieve a high PCE.
How does the best device of this work compare in terms of PCE with other devices with the same cell configuration (FTO/meso-ETL/MAPI/C) reported in the literature?
It’s a good suggestion. In the FTO/c-TiO2/m-TiO2/ZrO2/perovskite/carbon architecture-based PCSs, we didn’t adopt expensive noble metal as electrode and sprio-OMeTAD as HTL. Therefore, the cost of fabricating the devices is relatively low, which is more commercialization friendly. Besides, the use of carbon back electrodes makes it possible to product large-area PSCs [1]. However, without the metal electrode and HTL, the PCEs of the devices were relatively low.
According to Han’s reports [1-2], the PCE of the devices with the same cell configuration was about 13%, which was consistent with our report.
[1] Mei, A.; Li, X.; Liu, L.; et al. A hole-conductor-free, fully printable mesoscopic perovskite solar cell with high stability. Science 2014, 345, 295-298. [https://doi.org/10.1126/science.1254763]
[2] Pei, J.; Timothy, W.; Noel, W.; et al. Fully printable perovskite solar cells with highly-conductive, low-temperature, perovskite-compatible carbon electrode. Carbon 2018, 129, 830-836. [https://doi.org/10.1016/j.carbon.2017.09.008]
3. Line 108: Please explain why the ZrO2 layer is deposited on top of the TiO2 one.
It’s a good suggestion. In the FTO/c-TiO2/m-TiO2/ZrO2/perovskite/carbon architecture-based PCSs, ZrO2 was blocking the flow of photogenerated electrons to the back contact, preventing recombination with the holes from the perovskite at the back contact, and acted as a scaffold to host the perovskite absorber with TiO2 layer [1].
[1] Mei, A.; Li, X.; Liu, L.; et al. A hole-conductor-free, fully printable mesoscopic perovskite solar cell with high stability. Science 2014, 345, 295-298. [https://doi.org/10.1126/science.1254763]
4. Line 137: Please explain how the absorbance was measured. With or without integrating sphere? Using the relation Absorbance = 1 – Reflectance – Transmittance?
It’s a good suggestion. We have only measured the transmittance without integrating sphere, and not using the relation Absorbance = 1 – Reflectance – Transmittance. We could get the data of absorbance, calculated with Lambert-Beer law, directly from the UV-vis spectrophotometry (UV3600, Shimadzu, Japan).
5. Line 142: Please provide more details about the PV & EQE measurements (forward/reverse scan, scan rate, light/voltage bias…).
Your suggestion is very helpful and we have corrected our manuscript. The detailed corrections are as followed:
The photocurrent-voltage (J-V) curves were used to study the photoelectric properties of the PSCs, which were obtained from an electrochemical workstation (ZAHNER-elektrik GmbH & Co. KG, Germany) with a solar light simulator (Oriel Sol3A, Newport Corporation, USA) via scanning from -1.1V to short circuit at a scan rate of 150 mV/s under AM 1.5G irradiation (100 mW/cm2) in ambient air. Incident photon-to-electron conversion efficiency (IPCE, Newport Corporation, USA) curves were acquired to analyse the photoelectric current of the sample cells under dark circumstances in ambient air.
6. Figure 2: Do the authors have at hand a better TEM image showing more clearly than in (e) the Pt shell and the outer Au NPs? Please change “size” to “radius”.
Thanks for your suggestion and we have added detailed corrections in Figure 2. Due to the limitation of distinguishability of the TEM, we didn’t get a better TEM image.
7. Line 166: Do the authors mean “radius” or “diameter”?
Thanks for your suggestion and we have corrected our manuscript. The detailed corrections are as followed:
(approximately 5 nm in radius)
8. Figure 3: How was absorption measured? With or without integrating sphere? Using the relation Absorbance = 1 – Reflectance – Transmittance?
It’s a good suggestion. We have only measured the transmittance without integrating sphere, and not using the relation Absorbance = 1 – Reflectance – Transmittance. We could get the data of absorbance, calculated with Lambert-Beer law, directly from the UV-vis spectrophotometry (UV3600, Shimadzu, Japan).
9. Line 183: Please provide a more solid evidence to the fact that the 380 nm and 800 nm resonances correspond to the inner and outer Au NPs: simulations or reference.
It’s a good suggestion. We have done some optical simulations about Au@Pt@Au core-shell NPs based on FDTD Solutions. The picture is as followed:
Figure 1. The simulation picture of Au (15 nm) @Pt (3 nm) @Au core-shell NPs with different radiuses of outer Au NPs.
From the picture above, we could know that as the radius of the outer Au NPs increase from 5 nm to 9 nm, the extinction peak at around 600 nm increased clearly, while the extinction peak at around 380 nm increased a little. It could be implied that the extinction peak at around 600 nm was related with the outer Au NPs, while the peak at around 380 nm related with the Au@Pt.
10. Figure 5: This figure is misleading, because the drawing on the left has a different vertical scale as the SEM image on the right. Please provide a better representation.
Thanks for your suggestion and we have added detailed corrections in Figure 5.
11. Figure 6: Please explain how the absorbance spectra were measured. On devices without carbon coating? With or without integrating sphere? Using the relation Absorbance = 1 – Reflectance – Transmittance? If only the transmittance was measured please show the transmittance spectrum instead of absorbance.
Also please provide the spectra in %, not arbitrary units.
It’s a good suggestion. The spectra were measured without carbon back electrodes. We have only measured the transmittance without integrating sphere, and not using the relation Absorbance = 1 – Reflectance – Transmittance. We could get the data of absorbance, calculated with Lambert-Beer law, directly from the UV-vis spectrophotometry (UV3600, Shimadzu, Japan). Some numbers of the absorbance data, calculated with Lambert-Beer law, are greater than 1. Therefore, it is not suitable to provide the spectra in %.
12. Lines 222-226: Adding the NPs increases the device absorbance in the 300-550 nm range and not at all at longer wavelengths (figure 6a). In contrast, the plasmon resonances of the NPs extend from 300 to more than 700 nm (figure 3a). If the plasmon resonances extend up to more than 700 nm and the NPs have a significant contribution to the device absorbance, then one might expect an improvement of the device absorbance at longer wavelengths than 550 nm upon adding the NPs. Why is it not observed? Please explain.
It’s a good suggestion. The device absorbance in the 300-550 nm range was consistent with the absorbance of plasmonic NPs. However, the enhancement of device absorbance in the 550-700 nm was not obvious. According to the relation Extinction = Absorbance + Scattering, the extinction was related with both the absorbance and scattering. Therefore, this phenomenon might be related with the scattering, also might be related with the strong absorbance capability of perovskite material in the wavelength of 300-600 nm.
13. Line 231: “The NPs can display excellent scattering because of the outermost small Au spheres”. This is not trivial. Please provide a reference or simulations to support this fact.
It’s a good suggestion. As depicted in Figure 6b, the outermost small Au spheres could be good scatterer to increase the path length of light. We could get this point in many literatures [3-4].
[3] Lu, Z.; Pan, X.; Ma, Y.; et al. Plasmonic-enhanced perovskite solar cells using alloy popcorn nanoparticles. Communication 2015, 5, 11175. [https://doi.org/10.1039/c4ra16385k]
[4] Munkhbayar, B.; Thomas, J.; William, J.; et al. Plasmonic gold nanostars incorporated into high-efficiency perovskite solar cells. Chemsuschem 2017, 10, 3750-3753. [https://doi.org/10.1002/ente.v5.10]
14. Table 1: Is there a way to estimate the volume fraction of NPs for a given wt%? What matters in optics is the volume fraction of absorbers/scatterers. This is needed to assess how much the NPs can affect the device absorbance.
It’s a good suggestion. I think it is possible to estimate the volume fraction of NPs for a given wt%. We could roughly calculate the volume fraction of NPs according to the thickness of mesoporous TiO2 layer, the thickness of the whole PSC, and the given wt%.
15. Figure 8: This figure is unclear. Please modify it. For example, use the same representation as in the front image of this work: https://doi.org/10.1039/C9TA01744E. If possible, include also the data for the reference cell (no NPs) and the cells with other NP wt% (not only 1%).
It’s a good suggestion. We have added detailed corrections in Figure 8.
Figure 8. Box charts of photovoltaic arguments for devices incorporated with 0-2 wt.% Au@Pt@Au core-shell NPs under simulated AM 1.5G irradiation (100 mW cm-2).
16. Figure 9: Why does EQE increase over the whole 300-800 nm spectral range, while the absorbance in figure 6 only increases in the 300-550 nm range? This does not seem very consistent with the fact that only optical effects affect EQE & Jsc. Please explain. I also wonder if the NPs might have a positive effect on charge extraction in short-circuit conditions.
It’s a good suggestion. Indeed, the IPCE spectra was not very consistent with the absorbance spectra of the devices, but was consistent with the absorbance spectra of the plasmonic NPs. There might be some problems while the measurement of the absorbance spectra of the devices. However, the overall trend was consistent. Many literatures have reported that the NPs have a positive effect on charge extraction. We are sorry for that we don’t have enough condition to do relevant measurements.

Round 2
Reviewer 2 Report
The authors have made an attempt to address earlier comments, but there are still few issues which must be addressed more sincerely before the MS is accepted for publication.
Authors claim to have got this MS edited by professional English editors, yet the English is not up to the mark. Following sentences in the Abstract demonstrate poor level of English in the manuscript. "...... performed the highest efficiency;" , "....and 6.38% at short-circuit current density..." Authors were asked to study and discuss the changes in the ELECTRICAL properties of TiO2 as a result of the addition of AuPtAu Nps. Instead, authors carelessly responds by writing a oversimplied and too generic statement - "TiO2 nanoparticles played a significant role in the transmission of electrons, which were usually employed as the electron transport layer. TiO2 nanoparticles could not only scatter uv light, but also absorbs uv light, which provided strong uv shielding. There are significant quantum size effects in the TiO2 nanoparticles. Therefore, TiO2 nanoparticles showed up special photophysical and photochemical properties." What happens to the electrical conductivity of TiO2? Changes observed in electron and hole mobility's? Authors should calculate and include the hysterisis index for devices with and without the Nps and discuss the significance of any observed change, rather than just replying with a loose statement saying "In this work, we had observed slight hysteresis change with introduction of nanoparticles in TiO2."
Author Response
1. Authors claim to have got this MS edited by professional English editors, yet the English is not up to the mark. Following sentences in the Abstract demonstrate poor level of English in the manuscript. "...... performed the highest efficiency", "....and 6.38% at short-circuit current density...".
Thanks for your advice and the English writing of the manuscript has been carefully edited again.
2. Authors were asked to study and discuss the changes in the ELECTRICAL properties of TiO2 as a result of the addition of AuPtAu Nps. Instead, authors carelessly responds by writing a oversimplied and too generic statement - "TiO2 nanoparticles played a significant role in the transmission of electrons, which were usually employed as the electron transport layer. TiO2 nanoparticles could not only scatter uv light, but also absorbs uv light, which provided strong uv shielding. There are significant quantum size effects in the TiO2 nanoparticles. Therefore, TiO2 nanoparticles showed up special photophysical and photochemical properties." What happens to the electrical conductivity of TiO2? Changes observed in electron and hole mobility's?
It’s a good suggestion. We are so sorry for the careless respond and the oversimple statement. We have revised the related content. The detailed description are as follows:
According to previous literature [1], the electrical conductivity of TiO2 layer improved with the loading of plasmonic NPs, which was helpful to the charge extraction. And Yuan et al. [2] introduced Au NPs into electron transport layer, enhancing the conductivity of the electron transport layer through the injection of hot electron, subsequently obtaining devices of excellent performance. Authors ascribed the enhancement to the lower series resistance, the faster extraction rates of carriers, and the elevation of the Fermi level of the electron transport layer, with the incorporating of Au NPs.
[1] Kakavelakis, G.; Petridis, K.; Kymakis, E. Recent advances in plasmonic metal and rare-earth-element upconversion nanoparticle doped perovskite solar cells. J Mater Chem A 2017, 5, 21604. [https://doi.org/10.1039/c7ta05428a]
[2] Yuan, Z.; Wu, S.; Bai, Z.; et al. Hot-electron injection in a sandwiched TiOx-Au-TiOx structure for high-performance planar perovskite solar cells. Adv Energy Mater 2015, 5, 1500038. [https://doi.org/10.1002/aenm.201500038]
3. Authors should calculate and include the hysterisis index for devices with and without the Nps and discuss the significance of any observed change, rather than just replying with a loose statement saying "In this work, we had observed slight hysteresis change with introduction of nanoparticles in TiO2."
It’s a good suggestion. We are so sorry for the loose statement and we have calculated and include the hysteresis index for devices with and without the NPs. The detailed description are as follows:
The hysteresis index could be calculated by the following formula [1]:
Therefore, according to the formula, the hysteresis index with and without plasmonic NPs was 0.089 and 0.084, respectively. The result indicated that the hysteresis effect had changed little with the loading of plasmonic NPs in this architecture-based PCSs.
[1] Severin, H.; Nakita, N.; Henry, S. Hysteresis Index: A Figure without Merit for Quantifying Hysteresis in Perovskite Solar Cells. ACS Energy Lett 2018, 3, 2472-2476. [https://doi.org/10.1021/acsenergylett.8b01627]

Author Response
Thanks for your advice and the English writing of the manuscript has been carefully edited again.
